# Minocycline Attenuates Lipopolysaccharide-Induced Locomotor Deficit and Anxiety-like Behavior and Related Expression of the BDNF/CREB Protein in the Rat Medial Prefrontal Cortex (mPFC)

**DOI:** 10.3390/ijms232113474

**Published:** 2022-11-03

**Authors:** Entesar Yaseen Abdo Qaid, Zuraidah Abdullah, Rahimah Zakaria, Idris Long

**Affiliations:** 1School of Health Sciences, Universiti Sains Malaysia, Health Campus, Kota Bharu 16150, Kelantan, Malaysia; 2Department of Physiology, School of Medical Sciences, Universiti Sains Malaysia, Health Campus, Kota Bharu 16150, Kelantan, Malaysia

**Keywords:** lipopolysaccharide, minocycline, memantine, medial prefrontal cortex, locomotor deficit, anxiety-like behaviour

## Abstract

Neuroinflammation following lipopolysaccharide (LPS) administration induces locomotor deficits and anxiety-like behaviour. In this study, minocycline was compared to memantine, an NMDA receptor antagonist, for its effects on LPS-induced locomotor deficits and anxiety-like behaviour in rats. Adult male Sprague Dawley rats were administered either two different doses of minocycline (25 or 50 mg/kg/day, i.p.) or 10 mg/kg/day of memantine (i.p.) for 14 days four days prior to an LPS (5 mg/kg, i.p.) injection. Locomotor activity and anxiety-like behaviour were assessed using the open-field test (OFT). The phosphorylated tau protein level was measured using ELISA, while the expression and density of brain-derived neurotrophic factor (BDNF) and cAMP response element-binding (CREB) protein in the medial prefrontal cortex (mPFC) were measured using immunohistochemistry and Western blot, respectively. Minocycline treatment reduced locomotor deficits and anxiety-like behaviour associated with reduced phosphorylated tau protein levels, but it upregulated BDNF/CREB protein expressions in the mPFC in a comparable manner to memantine, with a higher dose of minocycline having better benefits. Minocycline treatment attenuated LPS-induced locomotor deficits and anxiety-like behaviour in rats and decreased phosphorylated tau protein levels, but it increased the expressions of the BDNF/CREB proteins in the mPFC.

## 1. Introduction

A previous clinical study involving healthy volunteers showed that exposure to endotoxins induced locomotion deficits, anxiety-like behaviour, and neuroinflammatory responses [1]. Anxiety-like behaviour is described as an individual’s reaction to anticipated real dangers that may disturb homeostasis. This reaction may include behavioural responses, such as avoiding the source of the threat, scanning, and inhibiting ongoing behaviour, as well as physiological responses, such as increases in heart rate and blood pressure. Anxiety-like behaviour (pathological anxiety) is described as excessive or inappropriate physiological and behavioural reactions [2]. In this study, the grooming, rearing, and frequency of visits to the centre square of rats were used as measures to assess anxiety-like behaviour.

Similarly, previous animal studies have shown that either an intraperitoneal or a cerebroventricular injection of lipopolysaccharide (LPS) can induce locomotion deficits and anxiety-like behaviour [3]. As a result, LPS has been widely used in animal models to investigate the underlying mechanisms of locomotion deficits and anxiety-like-behaviour-induced neuroinflammatory diseases [4,5,6]. LPS stimulates the inflammatory pathway, which activates the formation of phosphorylated tau proteins (neurofibrillary tangles) and the accumulation of amyloids [7,8]. Inflammatory cytokines influence the phosphorylation of the BDNF receptor (TrkB), thereby further interfering with BDNF signalling [9], and tau proteins at least partially mediate the downregulation of Aβ-induced brain-derived neurotrophic factor (BDNF) [10]. The downregulation of BDNF and TrkB expressions in the hippocampus and cortex might lead to behavioural defects of depression and anxiety [11]. The cAMP response element-binding (CREB) protein regulates the transcription and subsequent expression of BDNF [12], and it is also involved in locomotion and anxiety-like behaviour [13]. A previous study demonstrated that a decrease in the levels of hyperphosphorylated tau proteins in the synapse of C57BL/6 mice led to improvements in anxious rat behaviour after ketamine treatment, which indicates a strong relationship between hyperphosphorylated tau proteins and mild stress models comprising anxiety and locomotion deficits [14]. Therefore, targeting tau proteins and the BDNF/CREB pathway is a potential strategy for the prevention and treatment of neurobehavioural-disorder-induced neuroinflammatory diseases.

Memantine is an antagonist of the N-Methyl-D-Aspartate (NMDA) receptor subtype of the glutamate receptor used in the symptomatic treatment of Alzheimer’s disease for over 30 years. It has shown anti-inflammatory effects and neuroprotective effects in several experimental and clinical studies, including those on locomotor deficits and anxiety-like behaviour [15,16]. A study on the acute treatment of memantine in naive rats induced a dose-dependent decrease in immobility in a tail suspension test (TST) but had no effect on locomotor activity [17]. However, long-term (9 weeks) treatment with memantine can impair motor behaviour and induce anxiety-like behaviour in naive mice [18].

Minocycline (microglial inhibitor) has been an approved therapeutic drug for the treatment of bacterial infection for over 30 years [19]. It is a semi-synthetic, second-generation, tetracycline antibiotic that can cross the blood–brain barrier (BBB) into the cerebrospinal fluid as a result of its small molecular size (495 KDa) and high solubility [20]. Recently, the developing research has been focusing on its neuroprotective properties in in vivo and in vitro animal models, as well as in clinical studies [21,22]. Recent studies have shown that minocycline can improve synaptic transmission and integrity, as well as neurologic function via reductions in phosphorylated tau protein levels and the upregulation of BDNF/CREB signalling pathways in several animal models [23,24].

A previous study also found that minocycline had an antidepressant-like effect in naive rats in a forced swim test (FST) [25]. Another study discovered that infusing minocycline into the cerebral ventricle of naive rats does not trigger the activation of locomotion in the OFT. This suggests that minocycline’s antidepressant-like effect is not due to the enhanced locomotion in naive rats [26].

However, the protection of minocycline against LPS-induced locomotor deficits and anxiety-like behaviour remains unclear. Thus, this study was designed to evaluate the effects of minocycline in comparison to those of memantine, an NMDA antagonist, on locomotion and anxiety-like behaviour, phosphorylated tau protein levels, and BDNF and CREB protein expressions in the mPFC of LPS rats.

## 2. Results

### 2.1. Effects of Minocycline on Food Intake and Body Weight in LPS-Injected Rats

Figure 1 displays the changes in food intake and body weight throughout the experiment. Food intake and body weight were assessed on day 1 and day 16 of the experiment. The LPS rats exhibited a significantly lower mean body weight (*p* ˂ 0.05) and mean food intake (*p* ˂ 0.05) than the control group in the experiment, indicating that LPS induced weight loss and reduced food intake. Interestingly, the food intake of the LPS rats treated with minocycline 25 and 50 mg/kg and memantine were significantly higher (*p* ˂ 0.05) than that of the untreated LPS rats, indicating that minocycline and memantine could prevent LPS-induced reduced food intake. There was no significant difference in food intake between the two doses of minocycline- and memantine-treated LPS rats (*p* > 0.05). The mean body weight of the LPS rats treated with minocycline 50 mg/kg was significantly higher than that of the untreated LPS rats and the LPS rats treated with minocycline 25 mg/kg and memantine 10 mg/kg (*p* ˂ 0.05). There was no significant change in the mean body weight between the untreated LPS rats and the LPS rats treated with minocycline 25 mg/kg and memantine 10 mg/kg.

### 2.2. Effects of Minocycline on the Locomotor Deficits and Anxiety-like Behaviour in LPS-Injected Rats

The effects of minocycline on locomotor deficits and anxiety-like behaviour are shown in Figure 2. The total distance, speed, and number of line crossings were used to assess the locomotion of experimental rats. The results show that there was a significantly lower total distance, speed, and number of line crossings (*p* < 0.05) in the LPS-injected rats than in the control group. Unlike the untreated LPS rats, the LPS rats treated with minocycline and memantine revealed a significantly higher total distance, speed, and number of line crossings (*p* < 0.05) than the LPS group.

Anxiety-like behaviours were assessed by measuring the rearing and grooming frequency, time spent in the centre of the open field, and frequency of entries into the centre. There was a significant decrease in the rearing frequency, time spent in the centre of the open field, and frequency of entries into the centre (*p* < 0.05) and an increase in the grooming frequency (*p* < 0.05) in the LPS rats compared to the control. Minocycline at both doses and memantine significantly increased the rearing frequency, time spent in the centre of the open field, and frequency of entries into the centre (*p* < 0.05) and decreased the grooming frequency (*p* < 0.05) compared to the LPS rats.

Interestingly, minocycline 50 mg/kg exerted a significantly higher total distance, speed, number of line crossings, rearing frequency, time spent in the centre of the open field, and frequency of entries into the centre (*p* < 0.05) and a lower grooming frequency (*p* < 0.05) than minocycline 25 mg/kg and memantine 10 mg/kg. No significant differences between minocycline 25 mg/kg and memantine (*p* > 0.05) were observed.

### 2.3. Effects of Minocycline on the Expression of BDNF- and Phosphorylated CREB-Positive Cells in LPS-Injected Rats

Figure 3 displays the expression and quantification of BDNF- and phosphorylated CREB-positive cells. The LPS rats showed a more significant decrease in the number of BDNF- (*p* ˂ 0.05) and phosphorylated CREB-positive cells (*p* ˂ 0.05) in the mPFC tissues compared with the control group. The minocycline- and memantine-treated LPS rats exhibited a more significant increase in BDNF- (*p* < 0.05) and phosphorylated CREB-positive cells (*p* ˂ 0.05) compared to the LPS group. There were no significant differences between minocycline (50 mg/kg), minocycline (25 mg/kg), and memantine (10 mg/kg) in the number of BDNF- (*p* > 0.05) and phosphorylated CREB-positive cells (*p* > 0.05).

### 2.4. Effects of Minocycline on the Expression Levels of BDNF and Phosphorylated CREB Proteins in LPS-Injected Rats

Figure 4 demonstrates the protein densities and mean IDV values of the BDNF and phosphorylated CREB proteins. The mean IDV values of the BDNF (*p* < 0.001) and phosphorylated CREB proteins (*p* < 0.001) in the mPFC tissues of the LPS group were significantly decreased compared to the control group. The mean IDV values of the BDNF (*p* < 0.05) and phosphorylated CREB proteins (*p* < 0.05) in the LPS group treated with minocycline and memantine were significantly increased compared to the LPS group. The mean IDV values of the BDNF (*p* < 0.05) and phosphorylated CREB proteins (*p* < 0.05) in the LPS group treated with minocycline (50 mg/kg) were significantly higher than those of the minocycline- (25 mg/kg) and memantine- (10 mg/kg) treated LPS rats. There was no significant difference in the mean IDV values of the BDNF (*p* > 0.05) and phosphorylated CREB proteins (*p* > 0.05) between the minocycline (25 mg/kg) and memantine groups.

### 2.5. Effects of Minocycline on Phosphorylated Tau Protein Expression Level on LPS-Injected Rats

This study also showed significant differences in the mean expression level of the phosphorylated tau proteins among all experimental groups, and this is shown in Figure 5. The mean expression level of the phosphorylated tau proteins was significantly higher (*p* < 0.05) in the LPS-injected group than in the control group. The mean expression level of the phosphorylated tau proteins was significantly lower (*p* < 0.05) in the minocycline- and memantine-treated LPS groups than in the LPS group. However, minocycline 50 mg/kg significantly reduced the phosphorylated tau protein expression levels (*p* < 0.05) more than minocycline 25 mg/kg and memantine 10 mg/kg.

## 3. Discussion

This study demonstrated two main findings: (1) LPS induced locomotor deficits and anxiety-like behaviour, which was accompanied by a decreased food intake and body weight, elevated phosphorylated tau protein levels, and decreased expressions of BDNF and phosphorylated CREB proteins in the mPFC, and (2) both doses of minocycline significantly improved locomotor deficits and anxiety-like behaviour. This study further verified the anxiolytic effects of minocycline, and minocycline was found to reduce phosphorylated tau protein levels and enhanced BDNF and phosphorylated CREB protein expressions in the mPFC of LPS rats, exhibiting effects comparable to those of memantine.

Our study also revealed that the LPS rats presented a lower food intake and body weight throughout the experimental period than the control group, which is consistent with a previous study [27]. The aetiology of weight loss and reduced food intake in the LPS rat model appears to be multifactorial, and several hypotheses have been postulated to explain it. LPS is known to induce sickness behaviours and growth failure, which are manifested by reductions in activity, exploration, social interaction, and the consumption of food and drink; fever; protein loss; hypersomnia; and the activation of the hypothalamic–pituitary–adrenal (HPA) axis and sympathetic system [28]. The minocycline- (50 mg/kg) treated LPS groups showed a restoration of body weight gain that was comparable to that of the control group, suggesting that minocycline treatment ameliorated LPS-induced weight loss. Previous reports suggested that the restoration of body weight and food intake in minocycline-treated LPS rats was attributed to its anti-glial-cell-mediated neuroinflammation [27]. The inhibitory effect of minocycline on glial-cell-mediated neuroinflammation in our study [29] is in line with a previous study and could be the underlying pathway of minocycline protection against LPS-induced decreased food intake and body weight. Memantine has also been shown to protect against LPS-induced neuroinflammation by activating microglia and astrocytes and increasing glutamate release, particularly the NMDA-dependent signalling pathway. Previous research suggests that decreasing NMDA receptor activation with memantine in the early stages of neuroinflammation reduces microglia activation and neuroinflammatory cascades, which can be attributed to the body weight restoration, locomotor deficits, and anxiety-like behaviour in this study [30].

This study confirmed that intraperitoneal LPS (5 mg/kg) significantly induced locomotor deficits, as exhibited by the reduced mean total distance, speed, and line crossings, and anxiety-like behaviour, as shown by the reduced rearing, time spent in the central square, and frequency of visiting the central square and the increased grooming frequency. Similar findings have been observed in other studies, with LPS inducing locomotor deficits and anxiety-like behaviour in several animal models [31,32]. The LPS-induced locomotor deficits and anxiety-like behaviour were reversed by the administration of minocycline (25 and 50 mg/kg) and memantine (10 mg/kg) for 2 weeks, which is in line with other studies that showed minocycline improved locomotor activity and exerted anxiolytic effects in several animal models [22,33].

A previous study found that LPS-induced depression-like behaviour in mice by decreasing CREB and BDNF expressions in the prefrontal cortex (PFC) and hippocampus [4]. Minocycline treatment can protect methylphenidate- [34] and alcohol- [24] induced neurodegeneration in the rat brain by upregulating CREB and BDNF expressions. In a rat model of chronic, unpredictable, stress-induced depression, memantine treatment was shown to have an antidepressant-like effect by preventing hippocampus mitochondrial dysfunction and memory impairment via the upregulation of the CREB/BDNF signalling pathway [35].

Interference with neurotrophin expressions and functions in the frontal, parietal, temporal, and occipital cortices has been proposed as one of the underlying mechanisms of LPS-induced neurobehavioural impairment. A previous study demonstrated that LPS stimulates pro-inflammatory cytokine production, which inhibits brain BDNF expression via the activation of the hypothalamic–pituitary–adrenal (HPA) axis [36]. Additionally, several reports have proposed that proinflammatory cytokines stimulate amyloid deposition and neurofibrillary tangle formation, which mainly consist of hyperphosphorylated tau proteins that trigger inflammatory and oxidative stress pathways. The inflammatory cytokines interfere with BDNF signalling by reducing the phosphorylation of the BDNF receptor (TrkB) [37]. The downregulation of the BDNF/CREB signalling pathway leads to a disturbance in synaptic neurotransmission, an alteration in LTP, and subsequent neurobehavioural dysfunction [4,38,39]. These findings are in line with our findings that LPS increased phosphorylated tau protein levels and decreased BDNF and phosphorylated CREB protein expressions in mPFC, while minocycline and memantine treatment were able to reverse these changes by decreasing phosphorylated tau protein levels and increasing BDNF and phosphorylated CREB protein expressions.

Thus, to the best of our knowledge, this study shows for the first time a comparison between minocycline and memantine in the attenuation of locomotor deficits and anxiety-like behaviour, reductions in phosphorylated tau proteins, and the upregulation of BDNF and phosphorylated CREB protein expressions in the mPFC of LPS rats. This study postulated that minocycline reduces hyperphosphorylated tau protein levels, upregulates the BDNF/CREB signalling pathway in the mPFC, and ameliorates LPS-induced locomotor deficits and anxiety-like behaviour. Furthermore, the neuroprotective effects of minocycline observed in this study were dose-dependent, meaning that the higher the dose, the better the neuroprotective effects, even when compared to memantine.

This study’s limitations include its small sample size; the absence of a BDNF and/or CREB inhibition study; the effects of minocycline and memantine on naive animals; and the failure to assess beta-secretase (BACE) and C99 inflammatory protein levels, which could shed light on the relationship between neuroinflammation and amyloidosis. Future research should address these limitations.

## 4. Materials and Methods

### 4.1. Animals

Adult male Sprague Dawley rats were obtained from the Animal Research and Service Centre (ARASC), Universiti Sains Malaysia (USM). The rats were approximately 3 months old with a bodyweight of 270 ± 20 g. The rats were acclimatised to a new environment for one week prior to the start of the experiment. They were kept in polypropylene cages (32 × 24 × 16 cm) and had free access to a standard pellet rodent diet (Altromin, Lage, Germany) and tap water. The rats were exposed to 12 h light/dark cycles (lights off at 7 p.m. and lights on at 7 a.m.) and held at a 23 °C room temperature and 50 ± 5% relative humidity. The experimental protocol followed internationally accepted principles for laboratory animal use and care and was approved by the research and Ethics Committee of this university. The number of the Animal Ethics Approval is (USM/IACUC/2018/ (942) (114)).

### 4.2. Experimental Design

The experimental timeline is shown in Figure 6. The rats were randomly divided into five groups (n = 10) as follows: (i) control-treated with distilled water, (ii) LPS-treated with distilled water, (iii) LPS-treated with minocycline at 25 mg/kg [40], (iv) LPS-treated with minocycline at 50 mg/kg [40], and (v) LPS-treated with memantine at 10 mg/kg [41].

Minocycline (USP, 12601Twinbrook Pkwy, Rockville, MD, USA) and memantine (USP, 12601Twinbrook Pkwy, Rockville, MD, USA) were administered intraperitoneally once daily for 2 weeks from day 1 to day 14. LPS was obtained from *Escherichia coli* 0111:B4 (Sigma-Aldrich, St. Louis, MO, USA) and injected intraperitoneally once on day 5 at a dose of 5 mg/kg [16]. After 2 weeks of treatment, all rats were subjected to the OFT on day 15.

### 4.3. Open-Field Test (OFT)

The open-field test (OFT) was used to assess the locomotor activity and anxiety-like behaviour of the rats. The test was conducted between 8 a.m. and 12 p.m. The rats were brought into the behavioural room and acclimatised for 30 min prior to testing. After each trial, the animals were returned to their cages. The apparatus consisted of a Perspex cage (height = 27 cm, length = 90 cm, and width = 90 cm), and the bottom was divided into 25 small squares (16 × 16 cm). A video camera was placed 250 cm above the open field to record the trials (Arc Soft Total media 3.5, Lenovo, Petaling Jaya, Malaysia).

For testing, each rat was placed in the centre of the open-field arena, and locomotor activity was digitally recorded for 5 min. The open-field apparatus was wiped with 70% ethanol between trials and dried before the next trial to avoid smelling bias. Locomotion was assessed by measuring the total distance travelled in the open field, speed, and line crossings using Panlab Smart Video Tracking (Harvard Apparatus, Cambridge, MA, USA), and anxiety-like behaviour was assessed by manually calculating the rearing and grooming frequency, time spent in the centre of the open field, and frequency of entries into the centre. A schematic representation of rat movement is shown in Figure 7. After the behavioural assessment using the OFT, the rats were sacrificed by being deeply anaesthetised with an overdose of sodium pentobarbital (60 mg/kg body weight). The brain was immediately collected and divided into right and left hemispheres. Then, each right and left hemisphere was cut into 3 coronal sections (2.2 mm to 4.6 mm from the bregma) comprising the mPFC region. We randomly collected 1 section from each hemisphere, and for an immunohistochemistry analysis (one technique had a mixture of different sections in different positions), the brain tissues were preserved in 10% formalin.

### 4.4. Immunohistochemistry for BDNF- and Phosphorylated CREB-Positive Cell Expressions

The paraffin sections were dewaxed by immersion in xylene I and II solutions for 2 min each. After that, the slides were hydrated in decreasing dilutions of ethanol for 2 min each. For antigen (Ag) retrieval, the slides were placed in a pressure cooker containing Tris EDTA buffer at a temperature of 90 °C for 3 min, and after that, they were left to cool down inside dH_2_O for 2 min. The slides were placed in a sequenza immunostainer, a few drops of hydrogen peroxide (H_2_O_2_) blocking agent were added to the slides, and the slides were incubated for 5 min. Then, the slides were washed with dH_2_O for 2 min and immersed in Tris-buffered saline–Tween 20 (TBST) buffer twice for 5 min each. Primary antibodies (Santa Cruz; mouse BDNF and phosphorylated CREB; dilution = 1:200 and 1:100) were added, and the slides were incubated overnight at −4 °C. The slides were rinsed in TBST twice for 5 min each, and secondary antibodies (Santa Cruz, TX, USA; anti-mouse; dilution = 1:500 and 1:200) were added to the slides, followed by incubation for 1 h at room temperature. The slides were rinsed in TBST twice for 5 min each and flooded in 3, 3’-Diaminobenzidine (DAB) for 5–10 min at room temperature. Then, the slides were washed with dH_2_O for 2 min and dipped in haematoxylin for 5 s. Next, the slides were dehydrated in increasing dilutions of ethanol for 2 min each, immersed in xylene I and II for 2 min each, and mounted by placing coverslips onto the slides using cytoseal. The sections known to express the BDNF- and phosphorylated CREB-positive cells in the mPFC regions (2.2 mm to 4.6 mm from the bregma) were imaged under 40 and 100× magnification using an image analyser connected to a light microscope (Olympus Corporation, Tokyo, Japan). The BDNF- and phosphorylated CREB-positive cells in the mPFC region were counted using Image-J software http://imagej.nih.gov/ij, accessed on 1 February 2022). The counting was carried out within a 100 × 100 mm grid placed in the mPFC region using three random sections for each rat. Only clearly visible brown DAB cells were considered to be BDNF- and CREB-positive cells.

### 4.5. Western Blotting for BDNF and Phosphorylated CREB Protein Expression Levels

One coronal section from each right and left hemisphere was taken for a WB analysis. For the WB analysis, the proteins were extracted from the mPFC tissue using a radioimmunoprecipitation assay (RIPA) buffer. The homogenates were centrifuged at 12,000× g at −4 °C for 15 min (Hettich Zentrifugen, Tuttlingen, Germany). The protein quantifications in each supernatant were calculated using a Bradford protein assay kit (Bio-Rad, Hercules, CA, USA). The proteins (60 μg) after optimisation were denatured with sodium dodecyl sulphate (SDS) sample buffer and separated using 10% SDS-polyacrylamide gel electrophoresis (PAGE). The proteins were transferred to a polyvinylidene fluoride (PVDF) microporous membrane (Membrane Solutions, Auburn, WA, USA) and then blocked with 5% skim milk for 1 h at room temperature. The membrane was incubated with primary antibodies (Santa Cruz, TX, USA, mouse BDNF, phosphorylated CREB; dilution 1:500 each) overnight at −4 °C. β-actin was used as an internal standard, using mouse monoclonal anti-β-actin (1:500 dilution in Tris Buffer Saline (TBST), Santa Cruz, TX, USA) as the primary antibody. Secondary antibodies (Santa Cruz, TX, USA; anti-mouse; dilution = 1:5000) were added and incubated for 1 h at room temperature, and the protein bands on the membranes were detected using Clarity™ Western ECL substrate kits (Bio-Rad, Hercules, CA, USA).

The relative densities of the protein bands were evaluated using a densitometry using Fusion FX Chemiluminescence Imaging apparatus (Viber Lourmat, Eberhardzell, Germany) and quantified by Image J software (NIH, Madison, WI, USA). The mean relative intensity (by fold change) was measured using the following formula:Mean relative intensity = (IDV BDNF or phosphorylated CREB/IDV endogenous control) _targeted group_/(IDV BDNF or phosphorylated CREB/IDV endogenous control)_calibrator group_

The IDV endogenous control represents the values of β-actin, the targeted group represents the treatment groups, and the calibrator group represents the control group.

### 4.6. Phosphorylated Tau Protein Assay by Enzyme-Linked Immunosorbent Assay (ELISA)

For an ELISA analysis, the mPFC tissues from 1 coronal section of the left and right hemispheres were extracted quickly and placed in ice-cold saline. The mPFC brain tissues were weighed and homogenised (10% *w*/*v*) in ice-cold phosphate-buffered saline (PBS: 0.1 M, pH 7.4) to prevent enzyme degradation for 5 min. The homogenised tissues were centrifuged at 10,000× g for 10 min at −4 °C (Hettich Zentrifugen, Tuttlingen, Germany). The supernatants were allocated into Eppendorf tubes and preserved at −80 °C for the ELISA analysis.

The phosphorylated tau protein level in the mPFC tissue was quantified using a rat pτ ELISA kit according to the manufacturer’s instructions (Elabscience, Wuhan, China). Standards, blanks, and samples (100 μL each) were added to a 96-microwell plate. The microplate was sealed and incubated for 1 h and 30 min at 37 °C. The liquid was decanted from each well, and 100 μL of the biotinylated detection antibody working solution was added to each well. The microplate was sealed and incubated for 1 h at 37 °C. The solution was decanted from each well, and 350 μL of the wash buffer was added to each well. The solution was soaked for 1 min, aspirated from each well, and left to dry against clean absorbent paper (repeated 3 times). The horseradish peroxidase (HRP) conjugate working solution (100 μL) was added to each well, and the microplate was sealed and incubated for 30 min at 37 °C. The solution was decanted again from each well, and a repetition of the washing step 5 times was performed. A substrate reagent (100 μL) was added to each well, and the microplate was sealed and incubated for 15 min at 37 °C. Then, the stop solution (50 μL) was added to each well. The optical density (OD value) of each well was measured at a 450 nm wavelength on a microplate reader set (Thermo Fisher Scientific Inc., Waltham, MA, USA). Calculations were carried out by referring to the standard curve, and data are expressed as phosphorylated tau protein pg/m2L.

### 4.7. Statistical Analysis

The study data were analysed using Statistical Package for the Social Sciences (SPSS) software (SPSS Inc., Chicago, IL, USA) version 24, and they are presented as means ± standard errors of mean (SEM). Differences between groups were evaluated using a one-way analysis of variance (ANOVA) followed by the Bonferroni post hoc test. A probability value (*p*) of less than 0.05 was used to indicate significant differences.

## 5. Conclusions

Minocycline, in a dose-dependent manner, improved locomotor deficits and anxiety-like behaviour, decreased phosphorylated tau proteins, and upregulated BDNF/CREB protein expressions in the mPFC of LPS rats. A higher dose of minocycline showed better results than memantine. Thus, minocycline can be used as a preventive therapeutic drug for neuroinflammatory diseases.

## Figures and Tables

**Figure 1 ijms-23-13474-f001:**
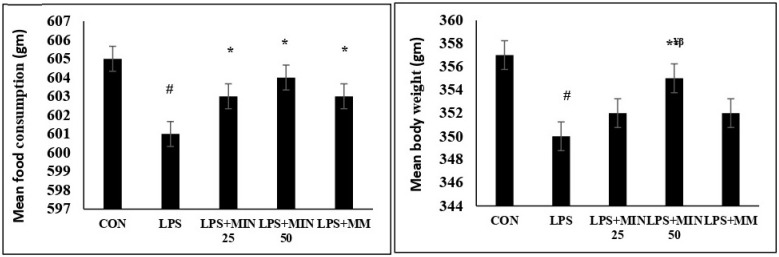
Mean food intake and body weight for all groups. One-way ANOVA test followed by Bonferroni post hoc test. Values are expressed as mean ± SEM. n = 10. (F (1, 36) = 6.34; ^#^
*p* < 0.05) versus control group; (F (1, 36) = 7.39; * *p* < 0.05) versus LPS group, (F (1, 36) = 12.10; ^¥^
*p* < 0.05) versus MIN 25, (F (1, 36) = 5.17; ^β^
*p* < 0.05) versus MM.

**Figure 2 ijms-23-13474-f002:**
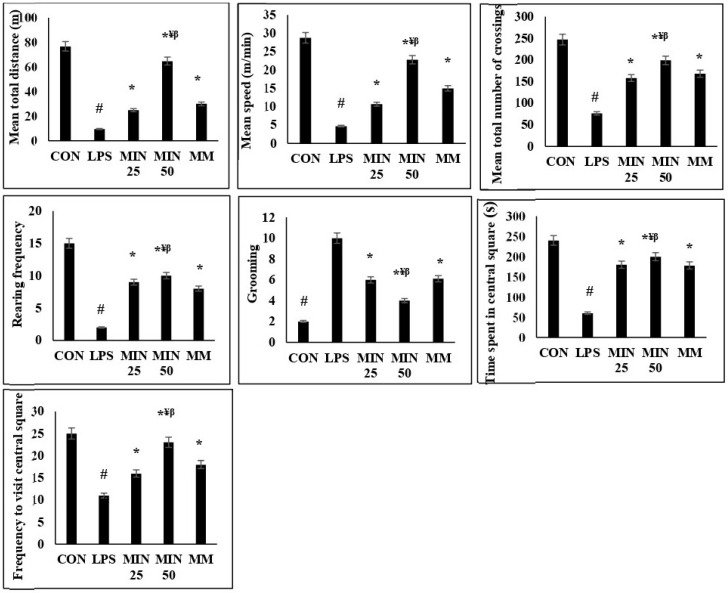
Mean of the total distance (1), speed (2), the number of line crossings (3), rearing (4), grooming (5), time spent in the central square (6), and frequency of visiting the central square (7) of all experimental groups during the open-field test. One-way ANOVA test followed by Bonferroni post hoc test. Values are expressed as mean ± SEM. n = 10. (F (1, 36) = 9.39; ^#^  *p* < 0.05) versus control group; (F (1, 36) = 4.13; * *p* < 0.05) versus LPS group, (F (1, 36) = 4.06; ^¥^  *p* < 0.05) versus MIN 25, (F (1, 36) = 7.55; ^β^  *p* < 0.05) versus MM.

**Figure 3 ijms-23-13474-f003:**
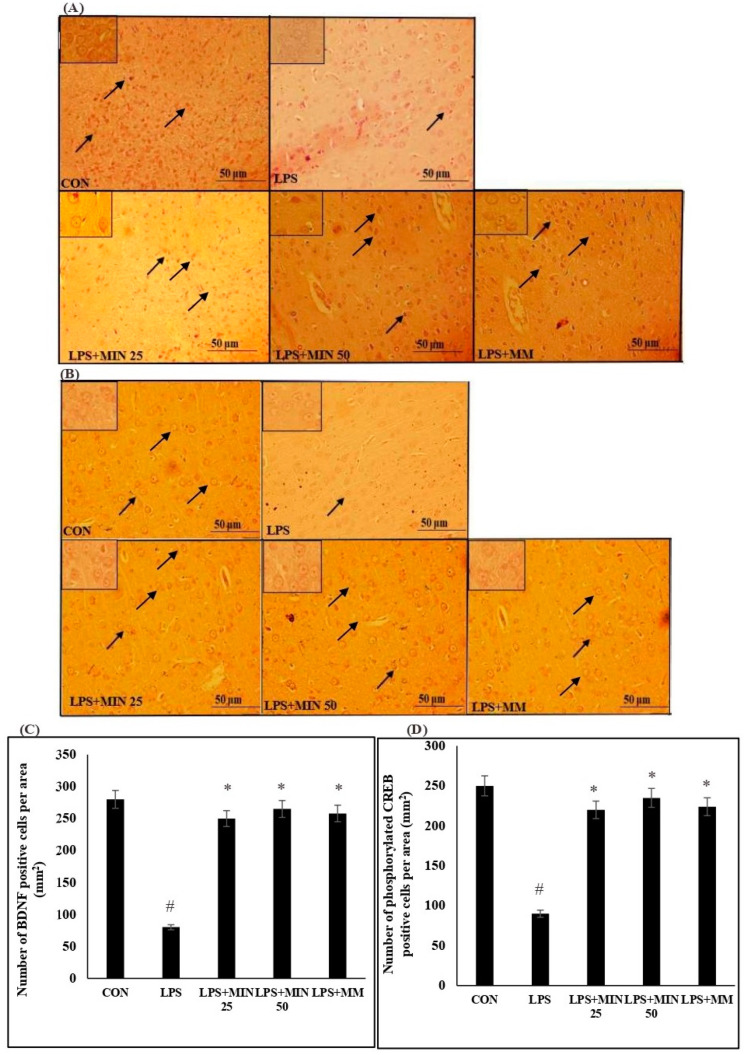
Distribution of BDNF- (**A**) and phosphorylated CREB-positive cells (**B**) in the mPFC at 40× and 100× magnification. The black arrows indicate BDNF- and phosphorylated CREB-positive cells. (**C**) Total number of BDNF- (**D**) phosphorylated CREB-positive cells in the mPFC. One-way ANOVA test followed by Bonferroni post hoc test. Values are expressed as mean ± SEM. n = 10. (F (1, 36) = 9.90; ^#^
*p* < 0.05) versus control group; (F (1, 36) = 9.90); * *p* < 0.05) versus LPS group.

**Figure 4 ijms-23-13474-f004:**
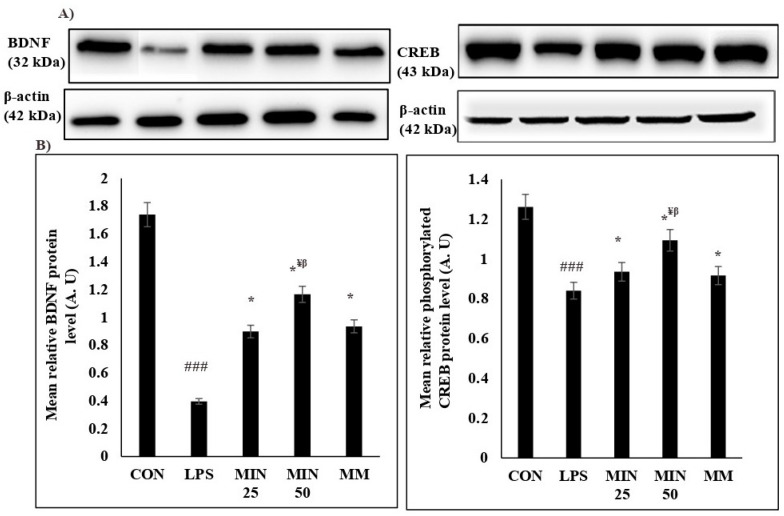
Effects of minocycline on the mean relative of BDNF and phosphorylated CREB protein levels in the mPFC. An example of Western blot results for all groups (**A**). The lower panel demonstrates the loading control. (**B**) Quantification analysis of IDV between the groups. The data were normalised by the control group. One-way ANOVA test followed by Bonferroni post hoc test. Values are expressed as mean ± SEM. n = 10. (F (1, 36) = 50.30; ^###^  *p* < 0.001) versus control group; (F (1, 36) = 11.38; * *p* < 0.05) versus LPS group, (F (1, 36) = 9.38; ^¥^
*p* < 0.05) versus MIN 25, (F (1, 36) = 7.86; ^β^
*p* < 0.05) versus MM.

**Figure 5 ijms-23-13474-f005:**
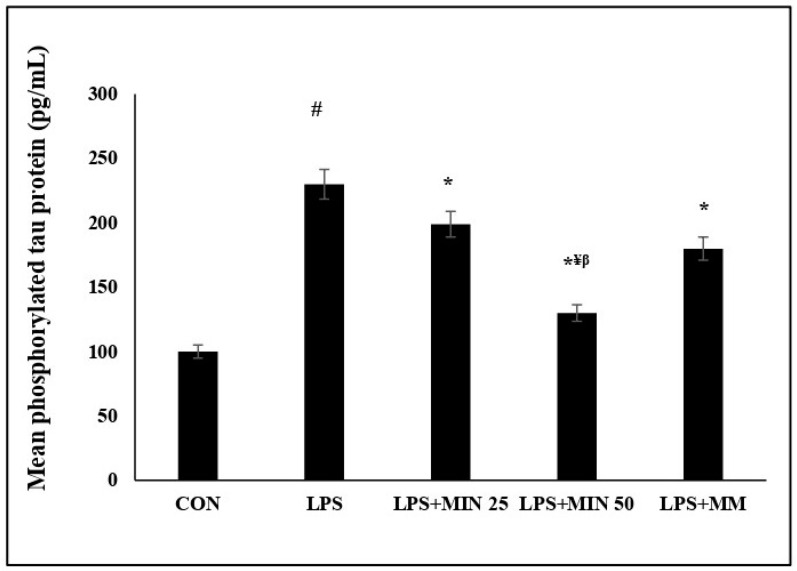
Effects of minocycline on the mean phosphorylated tau protein level in the mPFC. One-way ANOVA test followed by Bonferroni post hoc test. Values are expressed as mean ± SEM. n =10. (F (1, 36) = 25.2 and ^#^  *p* < 0.05) versus control group; (F (1, 36) = 4.59; * *p* < 0.05) versus LPS group, (F (1, 36) =7.39; ^¥^  *p* < 0.05) versus MIN 25, (F (1, 36) = 12.79; ^β^  *p* < 0.05) versus MM.

**Figure 6 ijms-23-13474-f006:**
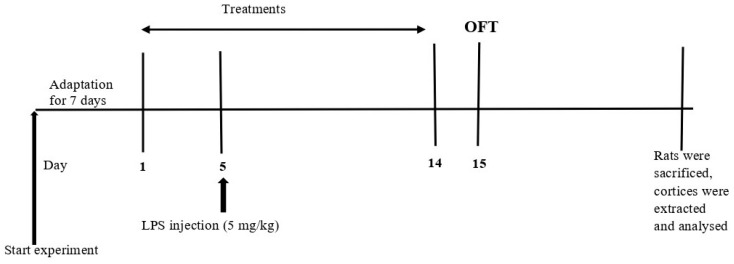
Experimental timeline.

**Figure 7 ijms-23-13474-f007:**
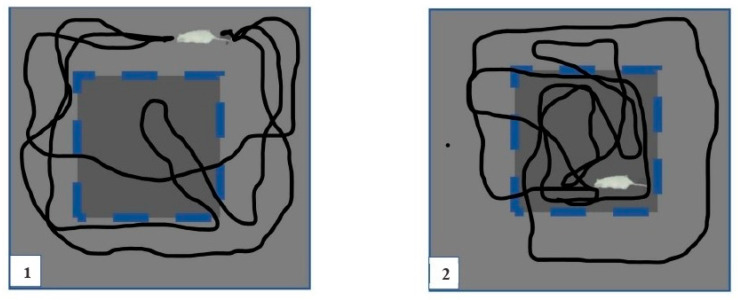
Rat activity in the open-field test (OFT). (**1**) Anxious rat’s behaviour in preference of peripheral arena. (**2**) Normal rat’s behaviour in preference of central arena.

## Data Availability

The datasets generated and/or analysed during the current study are available from the corresponding author on reasonable request.

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
