# Peer review of "Minocycline Attenuates Lipopolysaccharide-Induced Locomotor Deficit and Anxiety-like Behavior and Related Expression of the BDNF/CREB Protein in the Rat Medial Prefrontal Cortex (mPFC)"

_ijms, 2022, doi:10.3390/ijms232113474_

Round 1

Reviewer 1 Report

Please provide the details regarding the definition of anxiety-like behavioral. 

Numbers of subject are relatively small and should be noted in the limitation of the study and also provide the limitation of the study. 

Author Response

Please provide the details regarding the definition of anxiety-like behavioral

Anxiety is defined as the subject’s response to potential or real threats, which may induce impairment of homeostasis. The response may involve behavioural response such as avoidance of danger source, scanning and inhibition of ongoing behaviour and physiological response such as increase in heart rate and blood pressure. Excessive or maladaptive physiological and behavioural response is defined as anxiety-like behaviour (pathological anxiety)(Belzung & Griebel, 2001). In this study rat’s grooming, rearing and frequency to visit central square were considered parameters to evaluate anxiety like behaviour. Has been added in the text with highlighted red colour.

Numbers of subject are relatively small and should be noted in the limitation of the study and also provide the limitation of the study. 

Yes, agree with reviewer comments. Sample size of this study is small. In addition, this study did not assess BACE and C99 inflammatory protein levels which can elucidate the link between neuroinflammation and amyloidosis. Future studies are recommended to cover these limitations. The sentences regarding limitation of study has been added in the text.

Reviewer 2 Report

This manuscript presents the effect of two doses of minocycline and a single dose of memantine on the effects elicited by the systemic administration of lipopolysaccharide (LPS). Both minocycline and memantine revert the changes induced by LPS on locomotion, anxiety, BDNF and CREB protein expression, and tau phosphorylation. Although the results presented here are of interest, some of the findings are poorly discussed. Some major issues must be considered to be acceptable for publication.

Major comments:

-          In the title and throughout the manuscript it is indicated that the effect of minocycline on locomotion and anxiety is “via” the upregulation of BDNF/CREB “in the mFPC”. This asseveration goes beyond the experimental findings included in this manuscript. (1) The authors do not include a group with the BDNF and/or CREB inhibition to assure this link. (2) Minocycline is not locally administered in the mPFC, and no other brain areas (such as the hippocampus) have been studied.

-          It is not clearly stated why the authors study tau phosphorylation and its association with locomotor deficits and anxiety. Some behavioral parameters associated with learning/memory might be included. Moreover, it would be interesting to include the study of proinflammatory cytokines.

-          Do the authors have some data on the effect of minocycline and memantine in naïve animals? These groups should be included as some of the effects, mainly on anxiety, are opposite in naïve and experimental animals.

-          In the different sections of the Results, it is indicated the effect of MINO on food intake, body weight BDNF, etc. However, these experiments were not performed on naïve animals. It might be included that the observed effect was in LPS-treated animals.

-          The statistical analysis might be revised. In my opinion, there are some groups in which the statistical significance is not clear. For example, in the reversion of body weight in the LPS+MIN25 and LPS+MM compared to LPS; and in the CREB protein levels analyzed by WB in the LPS+MIN25 and LPS+MM compared to LPS. In addition, some of the statistical differences in the LPS group seem to be higher than indicated (only p<0,05 in the BDNF protein levels evaluated by WB?). Moreover, in my opinion, it would be easier to visualize the statistical differences using more than one symbol (*p<0,05, **p<0,01, ***p<0,001).

-          The analysis of the phosphorylated form of CREB would give a better idea of the pathway activation/inhibition in the LPS and LPS+MIN/MM groups.

-          How was the housekeeping analyzed in the CREB WB? The molecular weight of both proteins is 43 kDa for CREB and 42 kDa for b-actin, and they are evaluated using an ECL system.

-          The authors indicated that “LPS rats gradually reduced food intake…” however, the data presented here do not show a timeline. To prove this, it would be of interest to include the daily data on food consumption and body weight in figure 3, for the different experimental groups.

-          Why do the authors focus on mPFC? Other brain areas such as the hippocampus are also involved in anxiety and are more associated with tau hyperphosphorylation.

-          The effect of minocycline on BDNF and other parameters has been previously reported following LPS administration. Thus, the statement ‘… this study showed for the first time that….’ (line 338) is not correct. It might be more accurate to indicate that it is the first time that the effects of minocycline and memantine are compared.

-          What does ‘anti-tau protein property of minocycline’ (lines 341-342) means?

-          What is the proposed mechanism of action for memantine to reduce the LPS-mediated effect on body weight, food intake, and locomotion?

Minor comments:

-          Indicate the studies reporting the beneficial effects of minocycline on neuroprotection (line 56).

-          Include some information in the Introduction regarding memantine, and its effect on locomotion and/or anxiety.

-          Authors indicate that the OFT is performed from day 15 to day 18 (line 92). Were the animals tested more than once in the OFT? If not, indicate how were the animals divided to do the behavioral analysis. In my opinion, there is no need to include figure 2A showing the OFT arena. In figure 2B, it would be of interest to include what are the two experimental groups shown.

-          In the Material and Methods, it is indicated that the mPFC from the left hemisphere was used for both ELISA (line 164) and WB (line 147). The number of animals in the ELISA experiments is 10 per group. Clarify.

-          In my opinion, there is no need to include the pictures in Figure 5A. Just mention the atlas coordinates in the Material and Methods section.

-          Figures 5B and 5C might be fused to Figure 6. The images shown in Figures 5B and 5C are of very poor quality. Including a number of arrows parallel to the observed data, would help to interpret the image.

Author Response

In The title and throughout the manuscript it is indicated that the effect of minocycline on locomotion and anxiety is “via” the upregulation of BDNF/CREB “in the mFPC”. This asseveration goes beyond the experimental findings included in this manuscript. (1) The authors do not include a group with the BDNF and/or CREB inhibition to assure this link. (2) Minocycline is not locally administered in the mPFC, and no other brain areas (such as the hippocampus) have been studied.

Yes, agree with the reviewer. We do not include a group with the BDNF and/or CREB inhibition and locally administered of minocycline in the MPFC. Thus, we change our manuscript title accordingly to what we study.

However, we do have studied other areas of brain which is hippocampus but on learning and memory function and submitted elsewhere. (In press).

It is not clearly stated why the authors study tau phosphorylation and its association with locomotor deficits and anxiety. Some behavioral parameters associated with learning/memory might be included. Moreover, it would be interesting to include the study of proinflammatory cytokines.

1.     Previous study has demonstrated that a decrease in the levels of hyperphosphorylated Tau protein in synapse of C57BL/6 mice lead to improvement of anxious rat behaviour after ketamine treatment which indicate a strong relationship between hyperphosphorylated tau protein and mild stress model consisted  anxiety and locomotion deficit (Wen et al., 2019). This statement has been write in the introduction and highlighted with red colour.

2.     We also investigate the behavioral parameters associated with learning and memory and also measured the proinflammatory cytokines such as TNF-α, COX-2, Iba-1 and GFAP. However, the data were sent to another journal (unpublished).

Do the authors have some data on the effect of minocycline and memantine in naïve animals? These groups should be included as some of the effects, mainly on anxiety, are opposite in naïve and experimental animals.

Agree with the reviewer. Unfortunately, we don’t have the data on the minocycline and memantine in naïve animals. Due to cost constraint, we choose to concentrate on the effect of those drugs on animal that had been treated with LPS.

In the different sections of the Results, it is indicated the effect of MINO on food intake, body weight BDNF, etc. However, these experiments were not performed on naïve animals. It might be included that the observed effect was in LPS-treated animals.

Agree with the reviewer. Unfortunately, we don’t have the data on the minocycline and memantine in naïve animals. Due to cost constraint, we choose to concentrate on the effect of those drugs on animal that had been treated with LPS.

The statistical analysis might be revised. In my opinion, there are some groups in which the statistical significance is not clear. For example, in the reversion of body weight in the LPS+MIN25 and LPS+MM compared to LPS; and in the CREB protein levels analyzed by WB in the LPS+MIN25 and LPS+MM compared to LPS. In addition, some of the statistical differences in the LPS group seem to be higher than indicated (only p<0,05 in the BDNF protein levels evaluated by WB?). Moreover, in my opinion, it would be easier to visualize the statistical differences using more than one symbol (*p<0,05, **p<0,01, ***p<0,001).

Statistical analysis was revised. The changes were highlighted in red colour in the text.

The analysis of the phosphorylated form of CREB would give a better idea of the pathway activation/inhibition in the LPS and LPS+MIN/MM groups.

Yes. Actually, the protein that we used is a phosphorylated form of CREB. We change the CREB protein into phosphorylated CREB protein and highlighted the changes with red colour in the text.

 In How was the housekeeping analyzed in the CREB WB? The molecular weight of both proteins is 43 kDa for CREB and 42 kDa for b-actin, and they are evaluated using an ECL system.

Mean relative intensity (by fold of change) was measured by the formula:

Mean relative intensity = (IDV CREB/IDV endogenous control) targeted group / (IDV CREB/ IDV endogenous control) calibrator group.

IDV Endogenous control represents values of β-actin, the targeted group represents the Treatment groups, and the calibrator group stands for the control group. Add this formula on WB methods and highlight with red colour in red.

The authors indicated that “LPS rats gradually reduced food intake…” however, the data presented here do not show a timeline. To prove this, it would be of interest to include the daily data on food consumption and body weight in figure 3, for the different experimental groups.

Unfortunately, we don’t have the data for daily food intake.

Why do the authors focus on mPFC? Other brain areas such as the hippocampus are also involved in anxiety and are more associated with tau hyperphosphorylation.

We also investigate the brain area hippocampus. However, the data were sent to another journal (unpublished).

The effect of minocycline on BDNF and other parameters has been previously reported following LPS administration. Thus, the statement ‘… this study showed for the first time that….’ (line 338) is not correct. It might be more accurate to indicate that it is the first time that the effects of minocycline and memantine are compared.

Yes, agree with the reviewer. Changes has been highlighted in red colour in the text.

What does ‘anti-tau protein property of minocycline’ (lines 341-342) means?

This sentence has been changed in the text. Highlighted with red colour.

What is the proposed mechanism of action for memantine to reduce the LPS-mediated effect on body weight, food intake, and locomotion?

1.     It is N-methyl-D-aspartate antagonist and increases serotonin (5-HT) and dopamine in mice that lead to improvement of locomotion activity in mice (Onogi et al., 2009).

2.     LPS activates microglia and astrocytes, resulting in increased glutamate release, particularly N-methyl-d-aspartate (NMDA) -dependent signalling. Taken all together, previous study suggests that decreasing NMDA receptor activation using memantine in early stages of neuroinflammation leads to reduction of microglia activation and neuroinflammatory cascades (Rosi et al., 2006) which can be attributed to weight restoration, locomotor deficit and anxiety like behaviour effects of memantine in this study. The changes in the text were highlighted with red colour.

Indicate the studies reporting the beneficial effects of minocycline on neuroprotection (line 56).

The studies have been added. The changes were highlighted with red colour in the text.

Include some information in the Introduction regarding memantine, and its effect on locomotion and/or anxiety.

Has been added in the introduction and highlighted with red colour.

 Authors indicate that the OFT is performed from day 15 to day 18 (line 92). Were the animals tested more than once in the OFT? If not, indicate how were the animals divided to do the behavioral analysis. In my opinion, there is no need to include figure 2A showing the OFT arena. In figure 2B, it would be of interest to include what are the two experimental groups shown.

OFT was performed only once on day 15. Has been changed according to the reviewer comments.

In the Material and Methods, it is indicated that the mPFC from the left hemisphere was used for both ELISA (line 164) and WB (line 147). The number of animals in the ELISA experiments is 10 per group. Clarify.

The mPFC were divided into two halves (right and left). For ELISA we collect right half of mPFC of first group of rats (n = 5) and left half of mPFC of second group of rats (n = 5). For WB, we collect left half of mPFC of first group of rats (n = 5) and right half of mPFC of second group of rats (n = 5). So, for both ELISA & WB (n = 10)

In my opinion, there is no need to include the pictures in Figure 5A. Just mention the atlas coordinates in the Material and Methods section.

Has been changed according to reviewer suggestion.

Figures 5B and 5C might be fused to Figure 6. The images shown in Figures 5B and 5C are of very poor quality. Including a number of arrows parallel to the observed data, would help to interpret the image.

Has been changed according to reviewer suggestion.

Round 2

Reviewer 2 Report

The authors have answered some of my questions, but there are still some important points that in my opinion need to be clarified, not only to reply to my requirements as a reviewer but to make the manuscript clearer to the reader.

I include my first question and the authors reply to make it easier to understand. The new question is marked as ‘Reviewer_v2’ or ‘R_v2’.

Major comments:

Question 1:

Reviewer_v1: Do the authors have some data on the effect of minocycline and memantine in naïve animals? These groups should be included as some of the effects, mainly on anxiety, are opposite in naïve and experimental animals.

Authors: Agree with the reviewer. Unfortunately, we don’t have the data on the minocycline and memantine in naïve animals. Due to cost constraint, we choose to concentrate on the effect of those drugs on animal that had been treated with LPS.

R_v2: I do understand those reasons, however, the effect of minocycline and memantine on naïve animals has been reported previously by other authors. If not the experimental groups, at least add these references.

Question 2:

R_v1: In the different sections of the Results, it is indicated the effect of MINO on food intake, body weight BDNF, etc. However, these experiments were not performed on naïve animals. It might be included that the observed effect was in LPS-treated animals.

A: Agree with the reviewer. Unfortunately, we don’t have the data on the minocycline and memantine in naïve animals. Due to cost constraint, we choose to concentrate on the effect of those drugs on animal that had been treated with LPS.

R_v2: Sorry if my question was not clear. The point was that as minocycline and memantine treatments were performed only in LPS-treated animals (and not naïve), this might be indicated in each results subsection title. For example: “Effects of minocycline on food intake and body weight in LPS-treated animals”.

Question 3:

R_v1: The statistical analysis might be revised. In my opinion, there are some groups in which the statistical significance is not clear. For example, in the reversion of body weight in the LPS+MIN25 and LPS+MM compared to LPS; and in the CREB protein levels analyzed by WB in the LPS+MIN25 and LPS+MM compared to LPS. In addition, some of the statistical differences in the LPS group seem to be higher than indicated (only p<0,05 in the BDNF protein levels evaluated by WB?). Moreover, in my opinion, it would be easier to visualize the statistical differences using more than one symbol (*p<0,05, **p<0,01, ***p<0,001).

A: Statistical analysis was revised. The changes were highlighted in red colour in the text.

R_v2: Some of my initial concerns about the statistical analysis have been clarified. However, there are still some questions regarding this topic. Is the number of animals 10 for all the experimental groups in all the experiments? Authors might consider including the number of animals in the Figure captions. The F value of the one-way ANOVA might be also added.

Question 4:

R_v1: The analysis of the phosphorylated form of CREB would give a better idea of the pathway activation/inhibition in the LPS and LPS+MIN/MM groups.

A: Yes. Actually, the protein that we used is a phosphorylated form of CREB. We change the CREB protein into phosphorylated CREB protein and highlighted the changes with red colour in the text.

R_v2: Thank you for the correction. However, the total amount of CREB (non-phosphorylated CREB) should be also included in the Western blot experiments. This is of paramount importance as some chronic treatments lead to changes not only in the phosphorylated form but also in the total protein.

Question 5:

R_v1: How was the housekeeping analyzed in the CREB WB? The molecular weight of both proteins is 43 kDa for CREB and 42 kDa for b-actin, and they are evaluated using an ECL system.

A: Mean relative intensity (by fold of change) was measured by the formula:

Mean relative intensity = (IDV CREB/IDV endogenous control) targeted group / (IDV CREB/ IDV endogenous control) calibrator group.

IDV Endogenous control represents values of β-actin, the targeted group represents the Treatment groups, and the calibrator group stands for the control group. Add this formula on WB methods and highlight with red colour in red.

R_v2: Thank you for adding this information, which helps the reader to understand how was the analysis performed. However, my question remains unanswered. Sorry if it was not clear.

In the WB the bands were obtained by chemiluminescence. For the BDNF western blot, it is easy to discriminate between the BDNF bands and the b-acting ones (32 and 42 kDa, respectively). However, for the phospho-CREB WB, the target and the housekeeping sizes are very similar (43 and 42 kDa, respectively). How were those bands analyzed? Were the phospho-CREB and b-actin performed in different membranes?

Question 6:

R_v1: The authors indicated that “LPS rats gradually reduced food intake…” however, the data presented here do not show a timeline. To prove this, it would be of interest to include the daily data on food consumption and body weight in figure 3, for the different experimental groups.

A: Unfortunately, we don’t have the data for daily food intake.

R_v2: It would have been interesting to have these data, but I understand. However, it is still stated in the manuscript that LPS ‘gradually reduced food intake’ and that they ‘gained less body weight’ (page 10, Discussion section, second paragraph). It is clear that the food intake and the body weight were lower, but whether it was gradual or not can not be assured. Moreover, it has been reported that after an acute LPS administration there is a fast and dramatic loss of weight associated with the sickness behaviour, not a lower gain.

Question 7:

R_v1: What is the proposed mechanism of action for memantine to reduce the LPS-mediated effect on body weight, food intake, and locomotion?

A: 1.      It is N-methyl-D-aspartate antagonist and increases serotonin (5-HT) and dopamine in mice that lead to improvement of locomotion activity in mice (Onogi et al., 2009).

2.           LPS activates microglia and astrocytes, resulting in increased glutamate release, particularly N-methyl-d-aspartate (NMDA) -dependent signalling. Taken all together, previous study suggests that decreasing NMDA receptor activation using memantine in early stages of neuroinflammation leads to reduction of microglia activation and neuroinflammatory cascades (Rosi et al., 2006) which can be attributed to weight restoration, locomotor deficit and anxiety like behaviour effects of memantine in this study. The changes in the text were highlighted with red colour.

R_v2: Thank you for including this information. In my opinion, part of this explanation: ‘Memantine also has been shown…….. (Rosi et al., 2006)’, might be better moved to the Discussion section, at the end of the second paragraph, following the suggested mechanism of action of minocycline.

Minor comments:

Question 8:

R_v1: In the Material and Methods, it is indicated that the mPFC from the left hemisphere was used for both ELISA (line 164) and WB (line 147). The number of animals in the ELISA experiments is 10 per group. Clarify.

A: The mPFC were divided into two halves (right and left). For ELISA we collect right half of mPFC of first group of rats (n = 5) and left half of mPFC of second group of rats (n = 5). For WB, we collect left half of mPFC of first group of rats (n = 5) and right half of mPFC of second group of rats (n = 5). So, for both ELISA & WB (n = 10).

R_v2: Sorry, but I still do not understand this. As indicated at the beginning of page 4, the right hemisphere was preserved in 10% formalin for the immunohistochemistry studies. Then, the left hemisphere was also divided into right and left? What were the limits for the right and left parts?

Question 9:

R_v1: Figures 5B and 5C might be fused to Figure 6. The images shown in Figures 5B and 5C are of very poor quality. Including a number of arrows parallel to the observed data, would help to interpret the image.

A: Has been changed according to reviewer suggestion.

R_v2: I appreciate the changes. However, it is still not easy to see the positive cells. I consider that including in each experimental group a number of arrows parallel to the results shown in the graph, would help to interpret the images.

Additional questions:

-         -  In the Material and Methods section, in the WB and immunohistochemistry sections, the secondary antibodies are ‘anti-mouse’”; remove ‘BDNF and phospho-CREB’. Add the reference for the b-actin antibody in the WB section.

-         -  Include which is the area (mm2) evaluated for the immunohistochemical analysis.

-         -  In the Discussion section, second paragraph (page 11), it is indicated that ‘The minocycline and memantine treated LPS groups showed restoration of body weight gain that was comparable to the control group’. This is only right for MINO 50 mg/kg, but not for the lower dose of minocycline and memantine, as they do not significantly revert the reduction of body weight induced by LPS.

Author Response

Reviewer_v1: Do the authors have some data on the effect of minocycline and memantine in naïve animals? These groups should be included as some of the effects, mainly on anxiety, are opposite in naïve and experimental animals.

Authors: Agree with the reviewer. Unfortunately, we don’t have the data on the minocycline and memantine in naïve animals. Due to cost constraint, we choose to concentrate on the effect of those drugs on animal that had been treated with LPS.

R_v2: I do understand those reasons, however, the effect of minocycline and memantine on naïve animals has been reported previously by other authors. If not the experimental groups, at least add these references.

The effect of minocycline and memantine on naïve animals has been added with their references in discussion section.

R_v1: In the different sections of the Results, it is indicated the effect of MINO on food intake, body weight BDNF, etc. However, these experiments were not performed on naïve animals. It might be included that the observed effect was in LPS-treated animals.

A: Agree with the reviewer. Unfortunately, we don’t have the data on the minocycline and memantine in naïve animals. Due to cost constraint, we choose to concentrate on the effect of those drugs on animal that had been treated with LPS.

R_v2: Sorry if my question was not clear. The point was that as minocycline and memantine treatments were performed only in LPS-treated animals (and not naïve), this might be indicated in each results subsection title. For example: “Effects of minocycline on food intake and body weight in LPS-treated animals”.

Yes, the minocycline and memantine treatment were performed only in LPS-treated animal (and not naïve). Each results subsection was changes according to reviewer comments.

R_v1: The statistical analysis might be revised. In my opinion, there are some groups in which the statistical significance is not clear. For example, in the reversion of body weight in the LPS+MIN25 and LPS+MM compared to LPS; and in the CREB protein levels analyzed by WB in the LPS+MIN25 and LPS+MM compared to LPS. In addition, some of the statistical differences in the LPS group seem to be higher than indicated (only p<0,05 in the BDNF protein levels evaluated by WB?). Moreover, in my opinion, it would be easier to visualize the statistical differences using more than one symbol (*p<0,05, **p<0,01, ***p<0,001).

A: Statistical analysis was revised. The changes were highlighted in red colour in the text.

R_v2: Some of my initial concerns about the statistical analysis have been clarified. However, there are still some questions regarding this topic. Is the number of animals 10 for all the experimental groups in all the experiments? Authors might consider including the number of animals in the Figure captions. The F value of the one-way ANOVA might be also added.

The number of animals were n=10. The figure captions have been changed according to reviewer comments.

R_v1: The analysis of the phosphorylated form of CREB would give a better idea of the pathway activation/inhibition in the LPS and LPS+MIN/MM groups.

A: Yes. Actually, the protein that we used is a phosphorylated form of CREB. We change the CREB protein into phosphorylated CREB protein and highlighted the changes with red colour in the text.

R_v2: Thank you for the correction. However, the total amount of CREB (non-phosphorylated CREB) should be also included in the Western blot experiments. This is of paramount importance as some chronic treatments lead to changes not only in the phosphorylated form but also in the total protein.

Unfortunately, we only performed WB on the phosphorylated form of CREB but not on total CREB protein.

R_v1: How was the housekeeping analyzed in the CREB WB? The molecular weight of both proteins is 43 kDa for CREB and 42 kDa for b-actin, and they are evaluated using an ECL system.

A: Mean relative intensity (by fold of change) was measured by the formula:

Mean relative intensity = (IDV CREB/IDV endogenous control) targeted group / (IDV CREB/ IDV endogenous control) calibrator group.

IDV Endogenous control represents values of β-actin, the targeted group represents the Treatment groups, and the calibrator group stands for the control group. Add this formula on WB methods and highlight with red colour in red.

R_v2: Thank you for adding this information, which helps the reader to understand how was the analysis performed. However, my question remains unanswered. Sorry if it was not clear.

In the WB the bands were obtained by chemiluminescence. For the BDNF western blot, it is easy to discriminate between the BDNF bands and the b-acting ones (32 and 42 kDa, respectively). However, for the phospho-CREB WB, the target and the housekeeping sizes are very similar (43 and 42 kDa, respectively). How were those bands analyzed? Were the phospho-CREB and b-actin performed in different membranes?

We performed both beta actin and CREB in the same membrane during optimisation only. Then, we ran both at different membrane for analysis.

R_v1: The authors indicated that “LPS rats gradually reduced food intake…” however, the data presented here do not show a timeline. To prove this, it would be of interest to include the daily data on food consumption and body weight in figure 3, for the different experimental groups.

A: Unfortunately, we don’t have the data for daily food intake.

R_v2: It would have been interesting to have these data, but I understand. However, it is still stated in the manuscript that LPS ‘gradually reduced food intake’ and that they ‘gained less body weight’ (page 10, Discussion section, second paragraph). It is clear that the food intake and the body weight were lower, but whether it was gradual or not cannot be assured. Moreover, it has been reported that after an acute LPS administration there is a fast and dramatic loss of weight associated with the sickness behaviour, not a lower gain.

Agree with reviewer comments. We change the sentences according to reviewer comments.

R_v1: What is the proposed mechanism of action for memantine to reduce the LPS-mediated effect on body weight, food intake, and locomotion?

A: 1.      It is N-methyl-D-aspartate antagonist and increases serotonin (5-HT) and dopamine in mice that lead to improvement of locomotion activity in mice (Onogi et al., 2009).

2.           LPS activates microglia and astrocytes, resulting in increased glutamate release, particularly N-methyl-d-aspartate (NMDA) -dependent signalling. Taken all together, previous study suggests that decreasing NMDA receptor activation using memantine in early stages of neuroinflammation leads to reduction of microglia activation and neuroinflammatory cascades (Rosi et al., 2006) which can be attributed to weight restoration, locomotor deficit and anxiety like behaviour effects of memantine in this study. The changes in the text were highlighted with red colour.

R_v2: Thank you for including this information. In my opinion, part of this explanation: ‘Memantine also has been shown…….. (Rosi et al., 2006)’, might be better moved to the Discussion section, at the end of the second paragraph, following the suggested mechanism of action of minocycline.

Agree with the reviewer comments. We have cut this sentence and put in the discussion part second paragraph following suggested mechanism of minocycline.

R_v1: In the Material and Methods, it is indicated that the mPFC from the left hemisphere was used for both ELISA (line 164) and WB (line 147). The number of animals in the ELISA experiments is 10 per group. Clarify.

A: The mPFC were divided into two halves (right and left). For ELISA we collect right half of mPFC of first group of rats (n = 5) and left half of mPFC of second group of rats (n = 5). For WB, we collect left half of mPFC of first group of rats (n = 5) and right half of mPFC of second group of rats (n = 5). So, for both ELISA & WB (n = 10).

R_v2: Sorry, but I still do not understand this. As indicated at the beginning of page 4, the right hemisphere was preserved in 10% formalin for the immunohistochemistry studies. Then, the left hemisphere was also divided into right and left? What were the limits for the right and left parts?

The brain was immediately collected and divided into right and left hemispheres.  Then each of the right and left hemispheres were cut into 3 sections (2.2 mm to 4.6 mm from bregma) which are the mPFC region. Randomly we selected 1 section from each hemisphere for immunohistochemistry, WB and ELISA analysis. Thus, every analysis got sample from right and left hemisphere which taken by randomly.

Question 9:

R_v1: Figures 5B and 5C might be fused to Figure 6. The images shown in Figures 5B and 5C are of very poor quality. Including a number of arrows parallel to the observed data, would help to interpret the image.

A: Has been changed according to reviewer suggestion.

R_v2: I appreciate the changes. However, it is still not easy to see the positive cells. I consider that including in each experimental group a number of arrows parallel to the results shown in the graph, would help to interpret the images.

The number of arrows was changed parallel to the results shown in the graph as reviewer comments.

-  In the Material and Methods section, in the WB and immunohistochemistry sections, the secondary antibodies are ‘anti-mouse’”; remove ‘BDNF and phospho-CREB’. Add the reference for the b-actin antibody in the WB section.

The reference for b-actin antibody in the WB has been added.

-  Include which is the area (mm2) evaluated for the immunohistochemical analysis.

The area is (2.2 mm to 4.6 mm from bregma). We add this area on immunohistochemical analysis.

-  In the Discussion section, second paragraph (page 11), it is indicated that ‘The minocycline and memantine treated LPS groups showed restoration of body weight gain that was comparable to the control group’. This is only right for MINO 50 mg/kg, but not for the lower dose of minocycline and memantine, as they do not significantly revert the reduction of body weight induced by LPS.

Yes, agree with reviewer comments. We change the sentences according to reviewer comments.

Round 3

Reviewer 2 Report

The authors have answered most of my questions, but In my opinion there are still some parts that need to be clarified.

Major comments:

-          Regarding the total CREB levels, I might understand that the authors can not perform the required western blot. I understand, but please, include a sentence indicating if LPS, minocycline and/or memantine have been reported to modify that protein.

Minor comments:

-        -  Discussion section, second paragraph. In my opinion this new paragraph might be moved to the Introduction section, were minocycline and memantine are introduced, respectively.

-        - Discussion section, third paragraph. As indicated previously, in my opinion it cannot be considered that the reduction in body weight is “fast”, as the authors did not perform a time-course of that parameter. “Our study revealed that the LPS rats presented a lower food intake and body weight throughout the experimental period….”.

-        - Material and Methods section, before Immunohistochemistry. “Then each of the right and left hemispheres were cut into 3 sections (2.2 mm to 4.6 mm from bregma) which are the mPFC region. Randomly we collected 1 section from each hemisphere and for immunohistochemistry analysis, the brain tissues were preserved in 10% formalin.” I am sorry, but I still do not fully understand the procedure. Were the three sections coronal, sagittal or horizontal? The sections were ramdomly assigned to each technique (meaning that one technique had the same section, in the same position), or they were ramdomly collected (meaning that one technique had a mixture of different sections, in different positions)?

-        - Figure 5. Please, indicate in the Y axis of the graphs the number of positive cells per area (mm2).

Author Response

Reviewer 2

-Regarding the total CREB levels, I might understand that the authors cannot perform the required western blot. I understand, but please, include a sentence indicating if LPS, minocycline and/or memantine have been reported to modify that protein.

The sentence mention about LPS, minocycline and memantine effect on CREB protein has been added in discussion section as suggested by reviewer.

- Discussion section, second paragraph. In my opinion this new paragraph might be moved to the Introduction section, were minocycline and memantine are introduced, respectively.

The second paragraph in the discussion section has been moved to introduction sections where memantine and minocycline are introduced as suggested.

-Discussion section, third paragraph. As indicated previously, in my opinion it cannot be considered that the reduction in body weight is “fast”, as the authors did not perform a time-course of that parameter. “Our study revealed that the LPS rats presented a lower food intake and body weight throughout the experimental period….”.

The sentence has been changed according to reviewer suggested.

- Material and Methods section, before Immunohistochemistry. “Then each of the right and left hemispheres were cut into 3 sections (2.2 mm to 4.6 mm from bregma) which are the mPFC region. Randomly we collected 1 section from each hemisphere and for immunohistochemistry analysis, the brain tissues were preserved in 10% formalin.” I am sorry, but I still do not fully understand the procedure. Were the three sections coronal, sagittal or horizontal? The sections were ramdomly assigned to each technique (meaning that one technique had the same section, in the same position), or they were ramdomly collected (meaning that one technique had a mixture of different sections, in different positions)?

The section was cut via coronal sectioning. The section was collected randomly mean that one technique had a mixture of different sections, in different positions.

- Figure 5. Please, indicate in the Y axis of the graphs the number of positive cells per area (mm2).

The Y axis has been changed according to reviewer suggested.

Round 4

Reviewer 2 Report

All my questions have been answered.